# Enhancing Food Literacy and Food Security through School Gardening in Rural and Regional Communities

**DOI:** 10.3390/ijerph20186794

**Published:** 2023-09-21

**Authors:** Timothy P. Holloway, Sisitha Jayasinghe, Lisa Dalton, Michelle L. Kilpatrick, Roger Hughes, Kira A. E. Patterson, Robert Soward, Kylie Burgess, Nuala M. Byrne, Andrew P. Hills, Kiran D. K. Ahuja

**Affiliations:** 1School of Health Sciences, College of Health and Medicine, University of Tasmania, Launceston, TAS 7248, Australia; timothy.holloway@utas.edu.au (T.P.H.); sisitha.jayasinghe@utas.edu.au (S.J.); lisa.dalton@utas.edu.au (L.D.); robert.soward@utas.edu.au (R.S.); nuala.byrne@utas.edu.au (N.M.B.); andrew.hills@utas.edu.au (A.P.H.); 2Menzies Institute for Medical Research, University of Tasmania, Hobart, TAS 7000, Australia; michelle.kilpatrick@utas.edu.au; 3School of Heath Sciences, Swinburne University of Technology, Melbourne, VIC 3122, Australia; rmhughes@swin.edu.au; 4School of Education, College of Arts, Law and Education, University of Tasmania, Launceston, TAS 7248, Australia; kira.patterson@utas.edu.au; 5Burnie Works, Burnie, TAS 7320, Australia; melo@burnieworks.com.au

**Keywords:** school gardens, experiential learning, nutrition, child health, rural health, food literacy, food security

## Abstract

A qualitative case study approach with in-depth, semi-structured interviews of key school staff, and student feedback was used to assess a school kitchen and garden program in the regional area of North-West Tasmania, Australia. A detailed program description was produced to conduct a realist evaluation with a Context-Mechanism-Outcome configuration, followed by a program theory evaluation through the construction of a retrospective program logic model. Dedicated kitchen and garden spaces, knowledgeable teachers committed to the program, provision of sufficient materials and consumables, and support from the school and community were found to be the basic requirements to establish a program. Additionally, it is essential to integrate both the kitchen and garden teaching components into the school curriculum. The positive outcomes (e.g., engagement, participation, knowledge, skills, behavioral change) of the program were dependent on the underlying factors, including dedicated support of school leadership, teaching staff, and the parent body for effective student engagement in the teaching spaces and for wider engagement from families and the community. The students’ feedback provided supporting evidence of increased food literacy with improvements in their understanding, abilities, and attitudes towards gardening, producing healthy food, and preparing food. This may further lead to enhanced food security for students’ families and the broader community.

## 1. Introduction

Communities in rural and/or regional areas often have poorer health outcomes than their metropolitan counterparts, being challenged by geographic location and associated socio-economic determinants [1]. Consequently, these regions have a higher prevalence of chronic conditions, alongside higher rates of hospital admission, morbidity, and mortality associated with lifestyle related disease [1,2]. Preventative health approaches, including access to proper nutrition, would potentially ease the chronic disease burden, however, impacts on food security often make this difficult for many people. Food security can be defined as the state where all individuals have physical, social, and economic access to adequate, safe, nourishing, and culturally appropriate food that fulfills their dietary requirements and food preferences to lead a healthy and active life [3]. Food security is reliant on the maintenance of four pillars, including availability, access, utilization, and stability, and shortcomings in any of these components may render people subject to food insecurity [3]. Regrettably, food insecurity disproportionately affects population groups that may experience vulnerability and/or disadvantage, including people residing in rural and remote areas, potentially leading to limited access to nutritious food [4].

The resulting adverse diet-related health and well-being consequences may develop across all stages of life [5], and this often includes malnutrition, but also, paradoxically, obesity and associated development of chronic conditions, including cardiovascular disease and type 2 diabetes [6]. Plausible explanations for this contradiction include poor dietary quality and readily available energy dense, highly and/or ultra-processed foods [6,7], often predicated on poor understanding of food utilization, and the associated level of food literacy [8].

Food literacy is a set of inter-related knowledge, skills, and behaviors to meet food needs, including planning, selection, preparation and eating, which empower individuals, households, and communities to maintain and strengthen healthy diets [9]. Improving food literacy is essential for promoting food utilization and enhancing food security [10], which may ultimately lead to better health outcomes for communities. The issue of food security, including strengthening food utilization, may be better addressed if placed-based and community-driven approaches are employed [11].

Schools play a significant role in promoting food literacy among children [12], although interventions have been heterogeneous in nature, highlighting the need for more targeted approaches to address key competencies [13]. The implementation of school garden-based programs may be an effective approach in addressing food literacy [12,13] and issues of food insecurity. These types of programs have gained widespread popularity in developed countries, more commonly located in urban environments [14], as they support experiential learning opportunities [15,16] and promote positive health and well-being outcomes [17,18]. These include changes in dietary and nutritional knowledge and behaviors [19], and increased intake of fruit and vegetables among school children [20], while also tackling the increased prevalence of childhood obesity [21,22].

The positive effects on children’s health and wellbeing are generally context specific, contingent on a combination of underlying mechanisms working mutually [23]. School gardening programs offer opportunities to improve food literacy in children and involve parents or guardians in the process, which can lead to improved food literacy in families and the ability to influence long-term dietary behaviors [24]. This provides prospect for enhanced food security, extending into the family and wider community [25,26]. Although most research has focused on school gardening in urban settings, limited research has characterized their influence in rural and/or regional areas, particularly the impact on food literacy and food security.

The aim of this study was to use a case study approach to evaluate a primary school kitchen and garden program in North-West Tasmania and answer the research question: “Do school-based gardening programs have potential to enhance food literacy and food security in rural and/or regional communities?”.

## 2. Materials and Methods

### 2.1. Setting and Context

This case study is centered on the Kitchen & Garden program in a primary school. This program is part of a suite of support programs designed to supplement core learning areas [27]. The school of focus in this instance, is one of nine public primary schools in the Burnie Local Government Area, which is managed by the Tasmanian Government Department for Education, Children and Young People, with approximately 290 children aged 4 to 11 years enrolled in kindergarten to Grade 6 [28]. Students and their families reside in the school intake area that includes suburbs of low relative socio-economic advantage compared to the rest of the state of Tasmania, Australia [29,30].

The Burnie Local Area Government Area is situated in rural/regional North-West Tasmania, Australia, with an estimated population of 19,918 in 2021 [31]. The population experiences considerable impacts on food security with almost 6% of adults reporting food insecurity in 2019 [32]. Additionally, approximately 16% of adults reported insufficient moderate/vigorous physical activity and 77% reported insufficient muscle strengthening, respectively, as well as 59% and 87% not meeting the guidelines for fruit and vegetable consumption, respectively [32]. Furthermore, over 76% of the adults in this area are overweight and obese compared to 59% for the rest of Tasmania [32].

### 2.2. Study Design, Ethics, and Participants

This study was part of the CAPITOL (Critical Age Periods Impacting the Trajectory of Obesogenic Lifestyles) Project, managed by the School of Health Sciences, College of Health and Medicine, University of Tasmania, with research team members having extensive expertise in exercise and nutrition science, public health, and education. For the development of the case study, a qualitative research methodology was utilized, which involved three stages—data collection, data analysis, and verification and reporting (Figure 1). Data collection involved in-depth semi-structured interviews, observational site visit, and review of the materials provided by participants. The data were analyzed using various methods, including the Template for Intervention Description and Replication (TIDier) methodology [33], appraisal of student feedback surveys, realist evaluation, and program theory evaluation.

The study was approved by The University of Tasmania Human Research Ethics Committee (application number H0018654). A purposive sampling approach was used to recruit a small number of key individuals and, as we utilized a constructivist paradigm, this was sufficient to attain in-depth and information-rich data [34]. Accordingly, the school principal (with overall responsibility for the Kitchen & Garden program) and teacher coordinating the school garden program (with day-to-day responsibility for teaching activities in the garden and related administrative duties) were invited (via email followed by phone communication) to participate in the study. The participants (n = 2, both male) provided their verbal consent.

### 2.3. Stage 1: Data Collection

An interview guide was produced to explore key foci associated with the program (Table 1). The interview questions were designed to identify needs, goals, activities, assumptions, resources, outcomes, evaluations, lessons learnt, partnerships, and further intelligence on the program. The guide was provided to participants prior to the interview, allowing for in-depth reflection and preparation of responses. Audio-recorded interviews of 45 min duration were conducted face-to-face on school premises, also enabling on-site observational visits of kitchen and garden spaces. One member of the research team (TPH) followed the interview guide, with additional probing to elicit further information when required. Interview audio-recordings were replayed, with participants’ responses transcribed and summarized. Participants provided additional print materials related to the program, including responses to the student feedback survey. This four-question survey was implemented by the school to assess the value of the program in the form of the enjoyment, skill development, and sustainability of the program.

### 2.4. Stage 2: Data Analysis

The TIDier itemized checklist reporting methodology was used to analyze the data generated from the interviews, and together with review of the additional provided material, was used to produce a thorough depiction of the program. Additionally, surveys completed by students were appraised and summarized. Realist evaluation was then employed to establish a Context-Mechanism-Outcome configuration prior to the construction of a retrospective program logic model, enabling elucidation of the “theory of change”. To ensure accuracy, the analysis process involved triangulation for consensus, with three researchers independently reviewing program details, materials, appraisals, and evaluations.

The rationale for the above approach was based on the fact that many community-based health promotion initiatives, such as those in school settings, develop organically with clear local ownership and purported positive health and well-being outcomes, but without formal evaluation to substantiate the work [35]. Inherent complexity may pose evaluative challenges, requiring appropriate methodologies to help explain underlying phenomena contributing to outcomes, which may include theory-led methods, such as realist evaluation and program theory evaluation [36]. Realist evaluation places emphasis on context and understanding how a program may work, by asking “what works, for whom, and under what circumstances?”, facilitated through consideration of causal elements and production of a Context-Mechanism-Outcome configuration [36,37]. Program theory evaluation also strives to determine how and why a program may work, although it aims to gain an explicit understanding of the underlying assumptions, facilitated through the construction of a retrospective program logic model explaining the theory of change [36,38].

### 2.5. Stage 3: Verification and Reporting

To ensure the interpretation of data was correctly representative of the program, the program description and evaluation results were verified and confirmed with the participants (principal and responsible teacher) prior to finalizing the report.

## 3. Results

### 3.1. Kitchen and Garden Program Description

The development of a program description revealed that the rationale for the Kitchen & Garden initiative was to enhance student’s food literacy by improving knowledge and understanding of healthy eating, food cultivation, and scientific principles (Table 2). The program had various objectives, which ranged from developing skills in cooking and gardening, to cross-curricula linking between theoretical learning and practical application, and to encouraging practices of sustainability.

### 3.2. Appraisal of Student Survey Responses

The summary of the student’s feedback (n = 9) to the survey conducted by the school provides supporting evidence on the students’ enjoyment, engagement, development of skills, access to experiential learning, in addition to sustainability of the Kitchen & Garden (KG) program. This student feedback was also considered for the realist evaluation and program theory evaluation. Below are selected comments:

Enjoyment: “I enjoy participating in the KG program!”; “I love it so much”; “I do enjoy doing all of the gardening”,

Engagement: “I enjoy planting thing(s)/feeding the chickens. I like cooking it’s pretty fun in my opinion”; “I enjoy it because I’m engaging in KG program and I like getting my hands dirty”; “Because it teaches us ways to take care of the enviro(n)ment and take care of animals”,

Skill development and experiential learning: “Learning more about making things out of clay & getting better with painting”; “I get to learn more about plants and animals. Learning about the plants are helpful because I have a garden at home”; “I’m learning how to garden and cook when I get to a(n) adult I’ll know how”.

Sustainability: “It’s important for kids to have fun at school sometimes so they want to come to school”; “Because KG can help a lot in the future and if you have a garden at home KG can help a lot”; “All schools should have the KG program (be)cause they learn life skills”.

### 3.3. Realist Evaluation

The contextual elements identified encompassed a wide range, including recognizing the need for the program, emphasizing the significance of having centrally located and inter-connected kitchen and garden spaces, and providing a committed teacher and learning assistant (Table 3). The mechanistic elements identified were diverse and comprised of dedicated support from the school and community, fostering student engagement and enjoyment, and gaining recognition for the program’s value from parents and families. The outcomes that influence in the short-term, mid-term, and long-term were identified.

### 3.4. Program Theory Evaluation

The theory of change for this program was exemplified by constructing a retrospective logic model (Figure 2). This model outlined the critical resources required, such as spaces, staff, and materials/consumables, along with the strategies to be employed, such as securing funding, creating educational materials, and implementing promotional strategies. The outputs of these efforts included the provision of funding, teaching, and engaging in various activities, which, in turn, led to short-term, mid-term, and long-term outcomes.

## 4. Discussion

We adopted a case study methodology to explore the potential of a primary school kitchen and garden program in enhancing food literacy and food security among students and families in a rural/regional community. The realist evaluation approach revealed that acknowledging the contextual factors is fundamental for the program’s success. This includes providing exclusive kitchen and garden spaces, having knowledgeable and dedicated teaching staff, procuring materials and consumables through support from the school and community, and complementing the core learning by incorporating the program into the curriculum and facilitating cross-curricula connections. Furthermore, the key underlying elements essential to outcomes ranged from dedicated support from the school leadership team, teaching staff, and parent body to effectively engaging students in the kitchen and garden spaces, and to garnering strong engagement and participation from families and the wider community.

After the construction of a retrospective program logic model, the theory of change inherent to the Kitchen & Garden program became evident, with available resources and strategies influencing outputs and outcomes, while simultaneously acknowledging the underlying assumptions necessary for change. The evaluation results demonstrate the program’s potential for enhancing food literacy, and consequently, promoting food security for children, their families, and the broader local community. Moreover, as rural communities may be disproportionately affected by food insecurity [4], the program logic model can serve as a framework for designing school gardening interventions beyond this particular context, highlighting the potential for the generalizability and transferability of this approach.

The attainment of improved food literacy was evidenced by linking curriculum teaching to experiential learning activities. For example, students engaged and participated in various activities such as planting vegetables and fruits from seed, harvesting, and consuming the produce, all while developing a fundamental theoretical understanding of the processes involved. Additionally, students had opportunities to care for animals, including chickens, which facilitated their understanding of animal husbandry practices, while also fostering a deeper connection to the origins and utilization of food. Further, the project work involved constructing “bee hotels” to attract bees to the garden for plant pollination, which supported student’s comprehension of plant and animal life cycles. The efficacy of experiential teaching in enhancing food literacy was supported by student survey responses, demonstrating improved knowledge, skills, and attitudes towards gardening, animal care, and cooking. The benefits of linking nutrition education with experiential approaches in school-based gardens has been shown to positively influence dietary habits in students, with increased fruit and vegetable knowledge and behavioral change [41]. Similarly, multidisciplinary approaches in school-based garden interventions may be best placed to improve children’s nutritional knowledge, attitudes, and behaviors towards vegetables, particularly when parents and the community are involved [19]. Given that the main domains of food literacy are the ability to plan, select, prepare, and eat [9], the Kitchen & Garden program was able to develop a range of core competencies that would lead to improved food literacy (e.g., understanding healthy nutrition, origins of food, gardening/cooking skills). It would be appropriate for future research to formally measure food literacy outcomes using validated tools.

The school places great emphasis on fostering strong relationships with students, teachers, parents, and the wider community. This aligns with its mission to empower individuals and is reflected in various programs. Because students recognized the practical value of the initiative and acknowledged the benefits of gardening knowledge for home-based gardens, this could lead to improved food security for families. Studies exploring the role of school gardens in other rural areas have similarly found that children do have the ability to gain food-related knowledge, skills, and values, which supports sustained food security in the community [26]. School gardening programs have the potential to augment the health and well-being of families at various levels. This can occur through both meso-level proximal effects, such as having a greater presence of family at schools and increased parental involvement, and family-level proximal effects, such as increased parental knowledge in nutrition, leading to improved dietary behaviors for the family [14,17]. In low-income settings particularly, direct participation may be necessary to encourage parental engagement and foster changed dietary behaviors [24]. Opportunities for parents/guardians to volunteer in the kitchen and/or school garden, promoting sharing of meals with students and staff, and distributing take-home kitchen-based and/or garden-based activities, may help in enhancing food security.

To expand school-based programs in rural or regional areas, and improve food literacy and food security, exploring public–private partnerships would be crucial. Following this study, the primary school partnered with 24 Carrot Gardens, a local not-for-profit organization that works with schools and the wider community, together with local volunteer architects, to co-design bespoke school gardens [42]. In rural and/or regional communities, collaboration between schools can facilitate the transfer of knowledge and skills, resulting in the creation of new programs or the strengthening of existing initiatives. Furthermore, these communities often have well established social networks and unique opportunities that can be utilized to strengthen school gardening programs. For example, establishing farm-to-school linkages can enrich the student experience and improve food literacy by providing a better understanding of food systems [43]. Similarly, opportunities exist in the area to partner with local educational institutions [44,45], that would facilitate expertise and knowledge transfer, further strengthening such programs.

Although this study has provided supportive evidence for enhanced food literacy, as well as potential for enhanced food security, the findings may not be transferable in all contexts. For example, schools may have limited resources to facilitate dedicated kitchen and garden spaces, teachers, expertise, and time, to develop and maintain effective programs. Similarly, assumptions that the school leadership team, the school, and broader community will be fully engaged with such programs may not hold true in all circumstances. Ideally, program logic models should be developed early in planning and subsequently used to inform monitoring and evaluation. However, we adopted a retrospective approach, which may be viewed as unorthodox, but nevertheless provided critical insight into program components and the theory of change. Additionally, the retrospective development of logic model has previously been conducted in childhood obesity research [46]. Furthermore, within the scope of our study, we were unable to evaluate the impact of the program on families and the community, which would be an important research objective for future studies.

## 5. Conclusions

In summary, this paper presents a case study of a primary school kitchen and garden program, demonstrating enhanced food literacy, and in turn, the potential for enhanced food security in children, families, and, by extension, the community. Given that many schools in rural and regional areas have opportunities to incorporate school-based gardens, a capacity exists for similar programs to have positive community impact. As there is an urgent need across public health domains for early intervention to prevent childhood malnutrition including obesity and the expanding chronic disease burden in rural and regional areas, well-designed evidence-based school-based gardening interventions provide viable options to help address these major problems.

## Figures and Tables

**Figure 1 ijerph-20-06794-f001:**
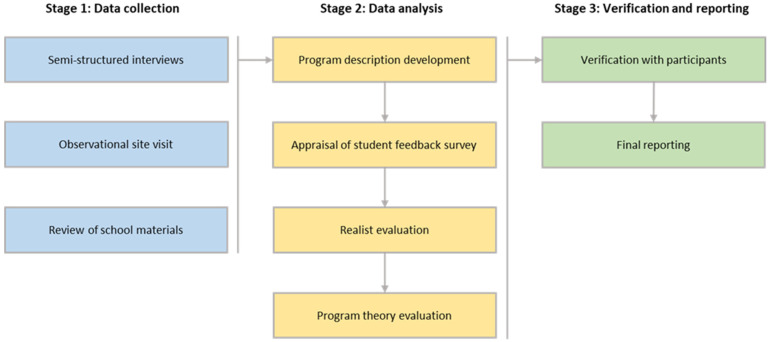
Stages of the research approach used for development of the case study to assess potential of the school Kitchen & Garden program in improving food literacy of children.

**Figure 2 ijerph-20-06794-f002:**
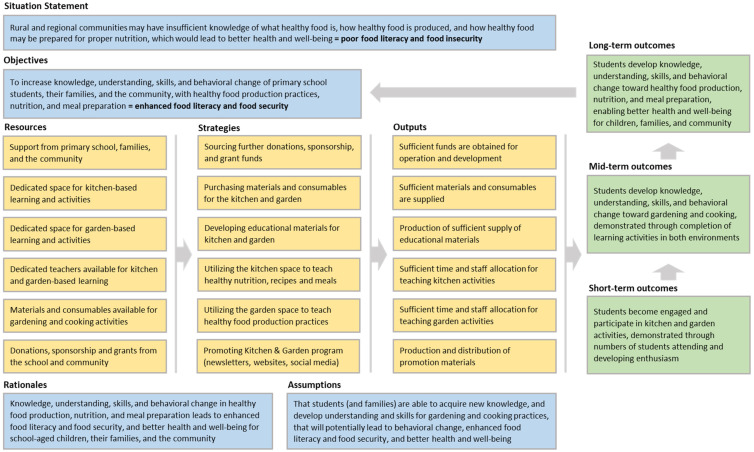
Retrospective program logic model of the school Kitchen & Garden program, exemplifying the “theory of change” with applicability to school gardening interventions in rural and/or regional areas.

**Table 1 ijerph-20-06794-t001:** Interview questions and associated foci utilized for collecting information on the school Kitchen & Garden program from the principal and main staff.

Question	Foci
1. Would you please explain why this program was developed?	Needs
2. What are the main objectives of this program?	Goals
3. Please describe the strategy components used for this program in lay terms?	Activities
4. What are the main assumptions in the strategy that enable the program to work?	Assumptions
5. What resources are required and available for this program?	Resources
6. What have been the main outcomes for this program?	Outcomes
7. Has this program been evaluated?	Evaluations
8. What are the main lessons that have been learnt from this program?	Lessons learnt
9. What partnerships are in place between this program and others in the community?	Partnerships
10. Would you be able to provide any additional information regarding the program?	Intelligence

**Table 2 ijerph-20-06794-t002:** Description of the school Kitchen & Garden program depicted in TIDier checklist format.

Item	Explanation	Description
Where?	Location of intervention	Primary school, Burnie, Tasmania, Australia (regional town in the State of Tasmania)
Why?	Rationale and theory	Based on need: anecdotal evidence indicated students experienced insufficient knowledge and understanding of origins, growing, harvesting, seasonality, and cycles of healthy food Program originally developed through the Stephanie Alexander Kitchen Garden Program (SAKGP), designed to provide a ‘seed to table’ experience for students [39,40]. However, the association was discontinued due to registration costs and strict adherence policies. Subsequently operated on informal basis, maturing into more established, formal, and embedded program
	Goal and objectives	To increase understanding of healthy eating, growing food, and scientific foundations of these processes, with specific objectives: o Provide skill development in gardening (e.g., planting, harvesting) and cooking (e.g., meal preparation) o Encourage good food hygiene practices, and safe use of gardening tools and kitchen utensils/appliances o Encourage healthy diets and nutrition choices among students o Gain an awareness of where food originates (e.g., “paddock to plate” philosophy) o Provide cross-curricula link between classroom learning and practical application o Gain an awareness and knowledge of responsible animal husbandry including good hygiene practices and proper nutrition o Encourage sustainability through recycling of food scraps, garden waste, and household items (e.g., utilizing worms farms, compost bins, and food for chickens) o Gain an understanding about life cycles of plants and animals (e.g., construction of “bee hotels” to encourage bees into the garden for plant pollination, supporting understanding of plant and animal life cycles) o Provide a safe and welcoming space for students who may have difficulty engaging with classroom activities
What?	Physical/informational material used in intervention	Kitchen spaces made available including utensils, appliances, and consumables Garden spaces centrally located including raised beds, garden workshop, potting shed, chicken house, fishpond, garden tools, and consumables
What?	Procedures and activities or processes used in intervention	Kitchen activities included recipe recording and meal preparation Garden activities included planting, harvesting, animal care, art, and carpentry projects, with seeds provided to students for growing at home-based gardens Activities provided cross curricula linkages (e.g., students tested moisture sensors in garden beds as an extension from classroom science lessons) Students received scrapbooks to record activities which were taken home at years end Students were able to transfer knowledge and skills learnt to home environment (e.g., gardening, cooking, and recycling) Ongoing partnerships with community members facilitated interest in the program to enable shared experiences (e.g., including The Lilium Society and Poultry Association)
Who?	Background and qualifications of providers	School principal had overall responsibility for program Teacher responsible for garden activities (0.6 FTE) Learning assistant responsible for kitchen activities (0.6 FTE)
How?	Mode of delivery	Conducted face-to-face in kitchen and garden spaces
When and how much?	Number of times and period of intervention delivery	Program conducted each week over three days, with nine classes of 45 min duration Program available to all students from Prep to Grade 4, and as an option for Grade 5–6 students in the Personal Pathways Program One third of students attended kitchen lessons and two thirds of students attended garden lessons Students attended at least three kitchen lessons and six garden lessons each school term Grade 5–6 students participating in the Personal Pathways Program attended in 90 min blocks for project focused work (e.g., including construction of planter and nesting boxes)
How well?	Evidence of intervention plans and development of program logic models	Evidence indicated program planning was on an ad hoc basis and intervention plans were not available No program logic models were developed for the program
Evaluation?	Evaluation details	Intervention was not formally evaluated, with evidence of impact mostly based on observational and anecdotal feedback provided by school staff, students, parents, and the community Feedback indicated program was successful due to strong commitment from school staff, including the leadership team, and strong community support through numerous donations, sponsorship, and grant funding (including local and national support)

Abbreviations: SAKGP, Stephanie Alexander Kitchen Garden Program; FTE, Full Time Equivalent.

**Table 3 ijerph-20-06794-t003:** Context-Mechanism-Outcome configuration of the school Kitchen & Garden program.

Context	Mechanism	Outcome
Primary school Kitchen & Garden program located in the Burnie Local Government Area of rural/regional North-West Tasmania Specific need identified by school community for greater knowledge, understanding and skills, in healthy diets and nutrition Incorporated into the teaching curriculum with regular student engagement in the kitchen and garden spaces Provision of one teacher and one learning assistant to deliver program activities Dedicated and centrally located kitchen and garden spaces with close physical connection Garden-based activities closely associated with kitchen-based activities Provision of materials and consumables made possible through school and community support	Dedicated support from the entire school community including principal, staff, and students Program important to the holistic development of students, with strong belief that objectives will be achieved Students enjoyed and engaged with accessible spaces provided and experiential activities delivered Program supports teamwork and collaboration between students leading to improved learning Parents and families saw value in how program contributes to student’s education Program reliant on dedication of key members of staff, including the principal, teacher coordinating garden activities, and learning assistant coordinating kitchen activities	Short-term outcomes: students are enthusiastic, engaged, and participate in the kitchen and garden activities, demonstrated through numbers of students attending and feedback Mid-term outcomes: students develop knowledge, understanding, skills, and behavioral change toward gardening and cooking, demonstrated through completion of learning activities in both environments Long-term outcomes: students develop knowledge, understanding, skills, and behavioral change toward healthy food production, nutrition, and meal preparation, enabling better health and well-being for children, families, and the community

## Data Availability

Not applicable.

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
