# Peer review of "Enhancing Food Literacy and Food Security through School Gardening in Rural and Regional Communities"

_ijerph, 2023, doi:10.3390/ijerph20186794_

Round 1
Reviewer 1 Report
This is a case study of a kitchen and garden programme in a rural school in North-West Tasmania providing a theory of change on the impact of the programme on food literacy. The authors use novel approaches of realist evaluation for complex interventions, however they could have utilised the opportunities that this method offers more. Below are a few specific recommendations.
1. A primary source of data are the in-depth interviews. However, the authors provide no details on their participants. Information like sample size, demographics, roles in the school and the intervention, recruitment processes, etc need to be reported.
2. The authors mention that they conducted validation of their initial logic model, but no further details are mentioned. What method was used for the validation process and who and how many people participated?
3. The authors seem to draw a lot from responses of students from a four-question survey implemented by the school to evaluate the programme. However, I have major doubts about the validity of this questionnaire. The questions are not presented in the manuscript, but they seem to have been put together before the logic model was developed. This means that they are not informed by it, although they are used to inform outcomes identified by the logic model. Responses are provided from 9 students (if all students in the school had access to the programme, this is a 3% response rate), so this further reduces reliability of the tool. I appreciate that responses to this survey can be considered together with other programme material, but I advise against using them to comment on the programme’s effectiveness in developing skills (e.g. page 9, line 14), knowledge transfer (e.g. page 9., line 17), and potential for transferability (e.g. page 9, line 22) in the manuscript, as well as in the abstract. It is not clear if any outcomes were also discussed in the in-depth interviews and what the findings were.
4. The CMOc and logic model lack valuable detail. For example:
a. Although, the research question of the paper refers specifically to programmes in rural/regional communities, the context section of the CMOc does not include any statements related to the fact that the intervention took place in a rural area.
b. Outcomes are not specific enough and the recommended tools for their assessment do not always correspond to the outcome. For example, the authors recommend that short-term outcomes of student engagement should be measured by number of students “developing enthusiasm”, but it is not clear how this can be measured. Recommendation to measure mid-term outcomes of awareness, knowledge, and skill development through completion of learning activities can also be questioned, as participation in activities does not necessarily lead to development of these attributes. No recommendation of how long-term outcomes can be measured has been provided.
c. Assumptions: The author’s assumption in the logic model is that in order to develop awareness, students should be able to develop awareness. I highly encourage authors to go further than that. How and under what circumstances can the students obtain this ability?
d. The CMOc and logic models focus on mechanisms that enabled the implementation of the programme but there is no discussion on barriers. Were these not discussed in interviews, and if so why?
5. In the discussion, the author’s make several claims about the programme’s effectiveness that do not seem to be based on specific outcomes measured as part of this work. This includes conclusions about the programme’s generalisability and transferability (e.g. page 12, line 20-23), attainment of improved food literacy (e.g. page 12, line 24), student’s improved knowledge, skills, and attitudes (page 12, line 34). I recommend rewording the discussion to only focus on measured outcomes. Similarly, in both the discussion (e.g. page 13, line 72) and the conclusion (page 13, line 82) the authors claim that the programme increased food literacy. I don’t think this represents the conclusion of this study, as food literacy was not explicitly measured. Authors should reword their conclusions to reflect findings from the work reported in this paper.
6. The retrospective nature of the program logic model development should be discussed as a limitation in the discussion.
Reviewer 2 Report
Comments are written in the text of the manuscript.
